# Development of Bionanocomposites Based on PLA, Collagen and AgNPs and Characterization of Their Stability and In Vitro Biocompatibility

**Maria Râpă** [1] , **Laura Mihaela Stefan** [2], **Traian Zaharescu** [3], **Ana-Maria Seciu** [2],
**Anca Andreea Țurcanu** [1], **Ecaterina Matei** [4,\*] , **Andra Mihaela Predescu** [4], **Iulian Antoniac** [4,\*]
and **Cristian Predescu** [4]

1   Center for Research and Eco - Metallurgical Expertise, University Politehnica of Bucharest,
    313 Spl. Independentei, 060042 Bucharest, Romania; rapa_m2002@yahoo.com (M.R.);
    andreea.tzurcanu90@gmail.com (A.A.Ț.)
2   National Institute of Research and Development for Biological Sciences, 296 Splaiul Independenţei, Sector 6,
    060031 Bucharest, Romania; lauramihaelastefan@yahoo.com (L.M.S.); ana.seciu@yahoo.com (A.-M.S.)
3   Radiochemistry Center, National Institute of Electrical Engineering (INCDIE ICPE CA), 313 Splaiul Unirii,
    P.O. Box 149, 060042 Bucharest, Romania; traian.zaharescu@icpe-ca.ro
4   Faculty of Material Sciences and Engineering, University Politehnica of Bucharest, 313 Splaiul Independentei,
    060042 Bucharest, Romania; andrapredescu@yahoo.com (A.M.P.); predescu@ecomet.pub.ro (C.P.)
\*  Correspondence: ecaterinamatei@yahoo.com (E.M.); antoniac.iulian@gmail.com (I.A.)

**Abstract:** Bionanocomposites including poly(lactic acid) (PLA), collagen, and silver nanoparticles (AgNPs) were prepared as biocompatible and stable films. Thermal properties of the PLA-based bionanocomposites indicated an increase in the crystallinity of PLA plasticized due to a small quantity of AgNPs. The results on the stability study indicate the promising contribution of the AgNPs on the durability of PLA-based bionanocomposites. In vitro biocompatibility conducted on the mouse fibroblast cell line NCTC, clone 929, using the 3-(4,5-dimethylthiazol-2-yl)-2,5-diphenyltetrazolium bromide (MTT) assay showed high values of cell viability (>80%) after cell cultivation in the presence of bionanocomposite formulations for 48 h, while the percentages of lactate dehydrogenase (LDH) released in the culture medium were reduced (<15%), indicating no damages of the cell membranes. In addition, cell cycle analysis assessed by flow cytometry indicated that all tested bionanocomposites did not affect cell proliferation and maintained the normal growth rate of cells. The obtained results recommend the potential use of PLA-based bionanocomposites for biomedical coatings.

**Keywords:** bionanocomposites; silver nanoparticles; biocompatibility; chemiluminescence; flow cytometry; cell morphology

## 1. Introduction

Medical device items based on polymers, metals, and ceramics are used for a variety of purposes, from diagnostic to therapeutic medical procedures. The use of nanocomposites in the medical field by application of nanotechnology reformed current medicine offering new opportunities to design products for different applications. Thus, a distinctive class of nanocomposites is based on natural and synthetic polymers as matrix combined with a nanofiller with mineral, organic, or metallic origins. The interactions between components provide important characteristics that certify the use of nanocomposites into medical devices [1]. Recently, the interest for poly(lactic acid) (PLA), a biodegradable aliphatic polyester, as a sustainable alternative for biomedical applications has been increased. Its favorable properties, such as biocompatibility, good mechanical strength, and stability

with respect to tissues, cells, enzymes, and various body fluids recommend it as one of the most important available biopolymer [2–4]. However, this biomaterial shows some barrier for medical areas related to its high rigidity, low toughness, and ductility. To overcome such weakness, PLA has been mixed with ecological plasticizers, such as polyethylene glycol (PEG) [5], acetyl tributyl citrate (ATBC) [6,7], and combination of ATBC and LAPOL 108 [8]. By introduction of nanofillers, such as ZnO nanoparticles [9], vinyl POSS (vinyl 50 polyhedral oligomeric silsequioxane) nanoparticles [10], carbon nanotubes [11], and nano-clay [12], into PLA matrix another useful feature like antimicrobial and UV-light screening properties are obtained. The use of PLA in material engineering is slowing down due to the bacterial adhesion [13]. To overcome these disadvantages, the usage of PLA coated with collagen, a synthetic tetracycline and minocycline (MH), and citrate-hydroxyapatite nanoparticles (cHA) was reported [13]. In another paper, Mania et al. [14] developed biocompatible and antimicrobial composites by use of chitosan into PLA. In [15], the authors immobilized glucosamine/chondroitin sulfate onto the PLA surface to obtain antibacterial PLA films bactericidal with increased cell viability.

Natural polymers such as chitosan, starch, collagen, and cellulose have presented a great interest, due to their biocompatibility and non-toxic properties. Also, it has been observed that natural polymers can favor cell attachment and proliferation, while synthetic polymers offer stability of the scaffolds [16]. Collagen is one of the most used natural polymers widely used for scaffold compositions being responsible for cell growth and cell penetration into the engineered matrix, as a biomimetic substitute of the skin. Collagen coated with poly(3-hydroxybutyric acid)–gelatin nanofibrous scaffold provides a soft bio-mimetic material for skin tissue engineering applications [17]. Kai et al. suggested the introduction of lignin into PLA to facilitate the control of oxidative stress caused by PLA formulations [18].

Silver is well recognized since antiquity for its antimicrobial activity against bacteria, fungi, and virus, nanotechnology being considered one of the most promising strategies to control microbial biofilms. On a large scale it is used as an ecological alternative for organic biocides and synthetic polymers. [19,20]. It was reported that the antibacterial activity of silver nanoparticles (AgNPs) is determined by their concentration, size, and shape [21]. High concentration of AgNPs led to both effective antimicrobial activity but potential toxicity in human health. Several alginate based nanocomposite films such as silver zeolite, silver nitrate, metallic silver, and citrate reduced AgNPs, and laser-ablated AgNPs were developed in order to apply these materials into medicine and food industry [22].

Various studies were developed using AgNPs with different polymers such as polyvinylpyrrolidone (PVP), poly(vinylalcohol) (PVA), hyperbranched polyurethane (PU), and poly acrylonitrile (PAN) in order to evaluate significant variations in the sizes and shapes of the Ag nanoparticles after interactions with mentioned polymers [23]. Biomaterials prepared by 3D printing of composites based on PLA loaded with Ag can be used extensively in different surgical treatment [24]. The enhancement of thermal stability of PLA was studied by the adding of nanohybrids as graphene oxide (GO)/AgNP [25].

In our previous works [5,6], PLA loaded with hydrolyzed collagen and AgNPs up to 10 wt.% and 1 wt.%, respectively, were melt processed and their biocompatibility and antifungal activities were determined with the attempt of their use as bionanocomposites for potential urinary drains. The present paper aims to study the effect of silver nanoparticles up to 1.5 wt.% on the surface (atomic force microscopy (AFM)), chemical structures (Attenuated Total Reflectance (ATR) Fourier Transform Infrared Spectroscopy (FT-IR)), thermal properties (differential scanning calorimetry (DSC) analysis) and thermal stability by chemiluminescence, as well as biocompatibility (in vitro cytotoxicity) of some PLA bionanocomposites prepared by melt mixing technique. This study could bring important knowledge about the biocompatibility and durability of some bionanocomposites prepared from PLA, collagen, and AgNPs with application as coating biomaterial.

## 2. Materials and Methods

### 2.1. Materials

The PLA pellets used in this study were Ingeo™ 2003D, as an alternative to conventional plastic, bought from NatureWorks LLC (Minnetonka, MN, USA). PLA shows the melting temperature ($T_m$) of 152.1 °C and vitreous transition temperature ($T_g$) of 59.9 °C (DSC; 10 °C/min; second heating scan); tributyl *O*-acetyl citrate (ATBC) was given by PROVIRON (Belgium) and used as a bioplasticizer having the density in the range of 1045–1055 kg/L; hydrolyzed collagen (Col) obtained by acid treatment of collagen from bovine tendon provided by the National Institute of Research and Development for Biological Sciences Bucharest was used as biocompatible agent; silver nanoparticles (AgNPs) (US Nanomaterials Research, INC., Houston, TX, USA) with particle sizes of 30–50 nm were used as antimicrobial agent. Additional chemicals used for in vitro biocompatibility assays were obtained from Sigma-Aldrich (Germany).

### 2.2. Preparation of PLA Bionanocomposites

Before processed, the powdered hydrolyzed collagen and PLA pellets were oven-dried under vacuum for 12 h at 40 °C and 80 °C, respectively. PLA-based bionanocomposites consisted of 5 wt.% hydrolyzed collagen and different ratios (0.5, 1.0, and 1.5 wt.%) of AgNPs were melt processing in a Brabender Plastograph at a temperature of 175 °C and rotor speed of 50 rot/min, for 8 min (Table 1). The bionanocomposites films necessary for further investigations were obtained by compression molding technology using a laboratory press at a pressing temperature of 175 °C. The PLA/ATBC blend was processed in the same conditions and used as reference.

**Table 1.** Tested poly(lactic acid) (PLA) bionanocomposites compositions.

| Sample | PLA (wt.%) | ATBC (wt.%) | Col (wt.%) | AgNPs (wt.%) |
|---|---|---|---|---|
| PLA/ATBC | 80 | 20 | - | - |
| PLA/Col5 | 76 | 19 | 5 | - |
| PLA/AgNPs 0.5% | 75.6 | 18.9 | 5 | 0.5 |
| PLA/AgNPs 1% | 75.2 | 18.8 | 5 | 1.0 |
| PLA/AgNPs 1.5% | 74.7 | 18.7 | 5 | 1.5 |

### 2.3. Methods

#### 2.3.1. Atomic Force Microscopy (AFM) Examination

A 4000SPM MultiView/NSOM atomic force microscope system (AFM) (Nanonics Imaging LTD) was used to investigate the surface morphology of the PLA bionanocomposites. SPM images were obtained using the non-contact mode (at a distance in the range of 0.1–10 nm of the sample). The analyzed surface area was of 80 μm$^2$ with an acquisition speed of 12 ms/point. Evaluation of the analysis was achieved by means of WSxM 4.0 software (Horcas, I; Fernandez, R., Gomez-Rodrigues, J.M.; Colchero, J.; Baro, A.M., Spain).

#### 2.3.2. Infrared Spectroscopy (ATR-FT-IR)

The functional groups presented in the bionanocomposites were identified by Fourier Transform Infrared Spectroscopy (FT-IR) using a FTLA 2000–104 spectrophotometer (ABB, Bomem Inc, Quebec, QC, Canada). Film samples were characterized using Attenuated Total Reflectance (ATR) using a Se-Zn crystal, at room temperature (25 °C), in transmittance mode. ATR-FT-IR spectra were registered in the spectral domain of 4000–750 cm$^{-1}$ with a resolution of 4 cm$^{-1}$.

### 2.3.3. Differential Scanning Calorimetry (DSC)

Thermal properties of bionanocomposites were evaluated using film samples (8 mg amount) with a DSC 823$^e$ calorimeter (Mettler Toledo, Greifensee, Switzerland). The scanning takes place from ambient temperature to 200 °C at a heating rate of 10 °C/min, was kept 2 min at 200 °C in order to erase the thermal history of the polymer and cooled down, and then heated again to 200 °C. Indium was employed as calibration standard for temperature and enthalpy ($T_m$ = 156.6 °C; $\Delta Hm$ = 28.45 J/g). Melting temperature ($T_m$) and its melting enthalpy ($\Delta Hm$) and glass transition temperature ($T_g$) were collected from the second heating scan for all blends. The degree of crystallinity ($X_c$ (%)) of the samples was established by the Equation (1):

$$X_c \% = (\Delta Hm / \Delta H_m^0 \times wt.\%) \times 100 \tag{1}$$

where $\Delta H_m^0$ is the theoretical enthalpy of fusion for a completely crystalline material and wt.% is the mass fraction of PLA. As cited from literature, $\Delta H_m^0$ for PLA is 93.1 J/g, which is assumed for 100% crystalline material [26].

### 2.3.4. Chemiluminescence Analysis

The thermal stability of PLA-based bionanocomposites was accomplished by isothermal and nonisothermal chemiluminescence (CL). The CL equipment, LUMIPOL 3 produced by Slovak Academy of Science, Bratislava, has a small temperature error (±0.5 °C). The polymeric samples with the weight around 3 mg were cut from pressed films and placed in aluminum pans. The emission intensity is divided by sample weight for avoiding the effect of the mass modification.

The studied samples were aged in accelerated regime by γ-exposure at room temperature in an irradiator (Ob Servo Sanguis, Budapest, Hungary) equipped with a $^{60}$Co source. The applied dose rate was 1 kGy h$^{-1}$.

### 2.3.5. In Vitro Cytotoxicity Tests

Mouse fibroblast cell line NCTC, clone L929, achieved from ECACC (Porton Down, Salisbury, UK) was used to evaluate in vitro cell cytotoxicity of the PLA bionanocomposite films by 3-(4,5-dimethylthiazol-2-yl)-2,5-diphenyltetrazolium bromide (MTT) and lactate dehydrogenase (LDH) assays. The bionanocomposites conditioned as films were cut in discs of 5 mm diameter and sterilized under UV light for 4 h. Cells were seeded at a density of $5 \times 10^4$ cells/mL in 24-well plates in Minimum Essential Medium (MEM) supplemented with 10% fetal bovine serum (FBS) (Biochrom, Berlin, Germany) and 1% antibiotics (penicillin, streptomycin and neomycin), and grown at 37 °C in humidified atmosphere with 5% $CO_2$. After 24 h, culture medium was replaced with fresh medium, samples were added into the wells (1 disk/well) and cells were further incubated in standard conditions for 24 and 48 h. At the end of these periods, the medium was discarded, cells were washed with phosphate buffer saline solution (PBS) and 500 μL of MTT working solution (0.25 mg/mL in MEM without FBS) was added to the cells and incubated for 3 h at 37 °C. The insoluble formazan crystals were dissolved with 500 μL isopropanol. After 15 min of incubation at room temperature with gentle stirring, the absorbance was quantified at 570 nm using a Mithras LB 940 microplate reader (Berthold Technologies, Bad Wildbad, Germany). The optical density registered directly corresponds to the number of metabolically active cells. The results were reported as the percentage of viability compared to the control sample (untreated cells) considered 100% viable. All samples were tested in triplicate per experiment and data were expressed as mean of three experiments ± SD.

LDH activity assay quantifies the amount of lactate dehydrogenase (LDH) delivered in the culture medium when cells are damaged or under stress and allows the evaluation of both cell membrane integrity and cell viability [27,28]. LDH activity was determined using CytoTox96 kit (Promega, Madison, WI, USA) according to the manufacturer's instructions. Briefly, after 24 h and 48 h of cells incubation in the presence of the bionanocomposites, enzymatic measurements of LDH released into

the culture medium were spectrophotometrically recorded at 490 nm using a 96-well plate reader Tecan Sunrise (Tecan, Grodig, Austria). Results were calculated using the following equation:

$$\text{LDH released (\%)} = (\text{LDH medium (OD490)} / \text{LDH positive control (OD490)}) \times 100 \qquad (2)$$

All samples were tested in triplicate per experiment, and the experiments were performed 3 times. The results were compared with the LDH positive control considered 100% cytotoxic, all samples showing low cytotoxicity percentages (below 20%), and correlated with those obtained by the MTT assay.

### 2.3.6. Cell Morphology

Cell morphology was examined after 48 h of cell incubation in the presence of the samples using hematoxylin-eosin staining. Light micrographs of NCTC clone 929 cells were obtained using a Zeiss Axio Observer D1 microscope and analyzed with AxioVision 4.6 software (Carl Zeiss, Oberkochen, Germany).

### 2.3.7. Cell Cycle Analysis by Flow Cytometry

Cell cycle analysis was assessed by flow cytometry, as previously described by Craciunescu et al. (2012) [29]. Briefly, Mouse fibroblast NCTC clone L929 cells were seeded in 6-well culture plates at a density of $2 \times 10^5$ cells/well in MEM culture medium supplemented with 10% FBS and incubated for 24 h at 37 °C in humidified atmosphere with 5% $CO_2$. PLA-based bionanocomposites were cut into discs of 10 m diameter and sterilized under UV light for 4 h. Cells were treated with PLA-based bionanocomposites (2 disks/well) and further incubated for 24 h. Then, the cells were trypsinized, washed with PBS and fixed with 70% ethanol overnight. Subsequently, cells were stained with 100 µg/mL RNase A (Promega, USA) for 30 min and with 50 µg/mL propidium iodide (PI - Becton Dickinson, Franklin Lakes, NJ, USA) for 30 min. The cellular cycle analysis was performed using a BD LSR II flow cytometer (Becton Dickinson, USA). Cell distribution (expressed in percentage) in each phase of the cell cycle (G0/G1, S, and G2/M) was analyzed with ModFit™ LT 3.0 software (Becton Dickinson, USA).

## 3. Results

### 3.1. AFM Examination

The morphology images for the plasticized PLA and PLA-based bionanocomposites surfaces are shown in Figure 1a–d.
Root mean square (RMS) calculated within the inspected areas is shown in Table 2.

**Table 2.** Roughness values for the PLA bionanocomposites and plasticized PLA estimated from AFM determination.

| Sample | Roughness, nm |
|---|---|
| Plasticized PLA | 34.4 |
| PLA/AgNPs 0.5% | 46.5 |
| PLA/AgNPs 1% | 49.3 |
| PLA/AgNPs 1.5% | 57.0 |

The morphology of the plasticized PLA (PLA/ATBC) showed a quite plate surface. A low rough surface of the plasticized PLA (Figure 1a) in comparison with those of bionanocomposites was observed. It is due to the interactions between polymeric matrices and plasticizer and the proper melt mixing conditions favoring the good dispersion of components. AFM analysis exhibited a "sea-island" morphology for the bionanocomposites containing collagen and silver nanoparticles. A relatively

greater roughness surface can be observed by the incorporation of the hydrolyzed collagen and silver nanoparticles in content of 0.5%, 1%, and 1.5% respectively (Figure 1b–d). This can be attributed to the dispersion of Col and AgNPs inside of the PLA/ATBC sample, which proved that the interface between the inorganic particles and PLA matrix is weaker.

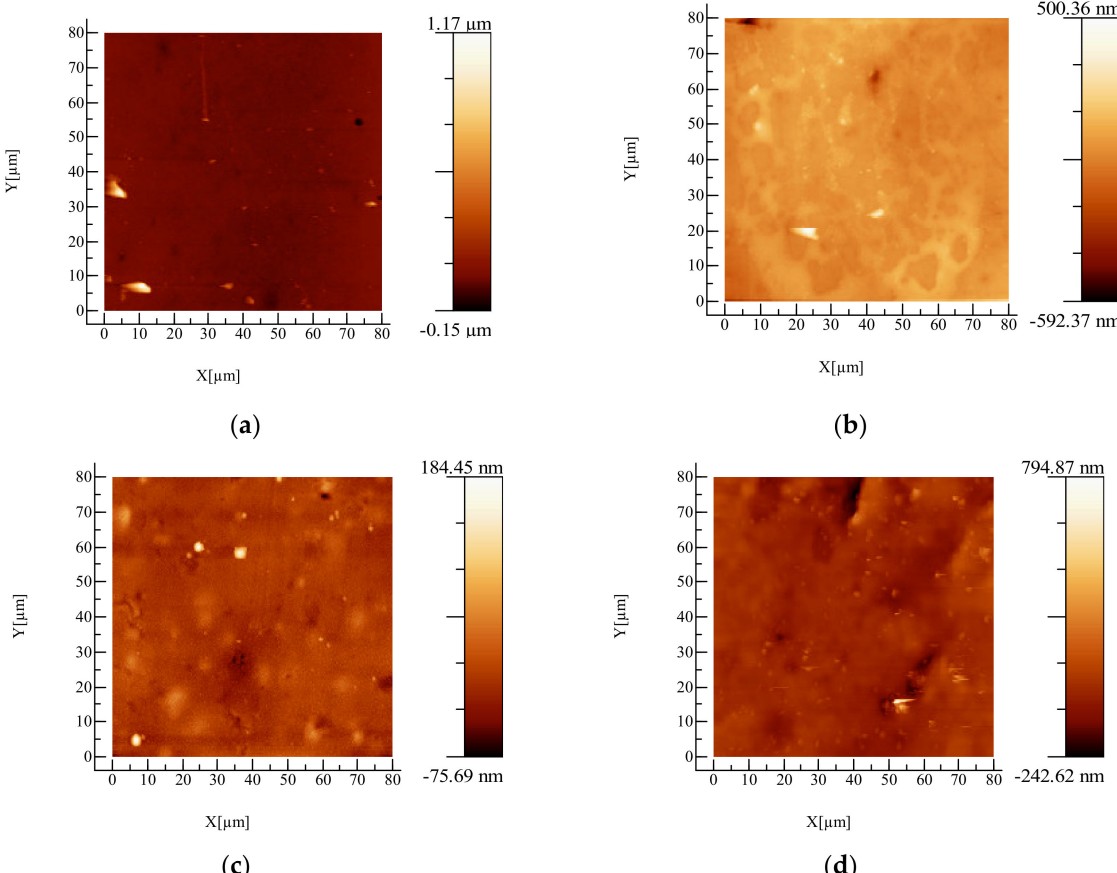

**Figure 1.** Atomic force microscopy (AFM) topography images: (**a**) Plasticized PLA; (**b**) PLA/silver nanoparticles (AgNPs) 0.5%; (**c**) PLA/AgNPs 1%; (**d**) PLA/AgNPs 1.5%.

### 3.2. Infrared Spectroscopy Analysis

The incorporation of hydrolyzed collagen and silver nanoparticles into plasticized PLA sample was analyzed by ATR FT-IR technique (Figure 2).

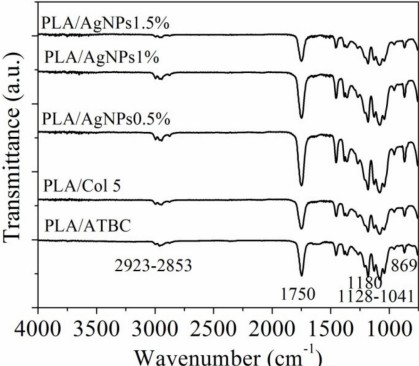

**Figure 2.** Normalized Attenuated Total Reflectance (ATR) Fourier Transform Infrared Spectroscopy (FT-IR) spectra of bionanocomposites containing PLA, Col, and AgNPs compared with that of plasticized PLA.

The spectral characteristic bands of bionanocomposites appeared around 1041, 1084, 1128, 1180, and 1750 cm$^{-1}$ are assigned to the motions of C–O stretching, C=O bending, similar to the main absorption peaks of typical PLA described in literature [30,31]. The introduction of Col 5 and AgNPs into mixtures generates some interactions that shift the pattern of typical bands of PLA/AgNPs blends. Thus, it is noticed that the carbonyl band shifts from 1751 cm$^{-1}$ for PLA/ATBC sample to 1747 cm$^{-1}$ for PLA/AgNPs 1.5% bionanocomposite, in good agreement with the increase in the degree of crystallinity (Table 3). It is noted, however, that the intensity of the absorption bands decreased as the AgNPs content increased in the blends; thus the amorphous band intensities at 869 cm$^{-1}$ (C–O–C) and 756 cm$^{-1}$ (C-H) decreased for PLA/AgNPs compared with those for PLA/ATBC spectrum.

### 3.3. Differential Scanning Calorimetry (DSC)

The thermal properties by DSC of PLA bionanocomposites films are presented in Figure 3.

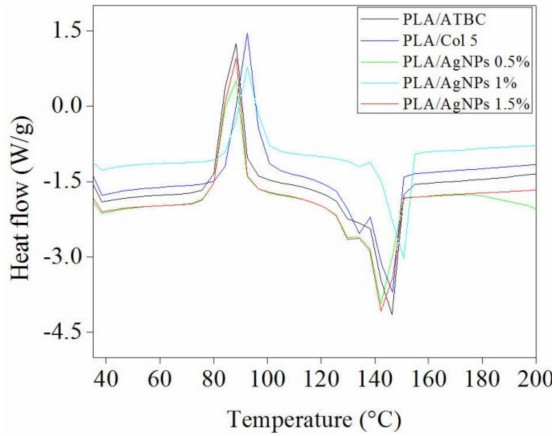

**Figure 3.** Normalized Differential Scanning Calorimetry (DSC) curves recorded in the second heating of PLA bionanocomposites films and plasticized PLA.

The glass transition temperature ($T_g$), crystallization temperature (Tc), melting temperature ($T_m$), and associated enthalpies of samples are evaluated from DSC curves and are summarized in Table 3.

**Table 3.** Thermal parameters evaluated from DSC thermograms and second heating scan.

| Sample | $T_g$ (°C) | $T_c$ (°C) | $\Delta H_c$ (J/g) | $T_m$ (°C) | $\Delta H_m$ (J/g) | $X_{c, PLA}$ (%) |
|---|---|---|---|---|---|---|
| PLA/ATBC | 42.3 | 86.7 | 20.5 | 145.8 <br> 131.7 | 22.1 | 29.6 |
| PLA/Col 5 | 46.5 | 92.0 | 22.7 | 145.50 <br> 133.23 | 21.2 | 29.9 |
| PLA/AgNPs 0.5% | 47.2 | 86.7 | 20.3 | 143.9 <br> 131.0 | 22.9 | 32.5 |
| PLA/AgNPs 1% | 48.0 | 92.5 | 22.9 | 150.8 <br> 134.3 | 23.8 | 33.9 |
| PLA/AgNPs 1.5% | 51.3 | 86.8 | 18.1 | 144.3 <br> 131.0 | 23.9 | 34.3 |

The thermal parameters of neat PLA evaluated from the DSC, second heating scan consist in $T_g$ = 59.9 °C and $T_m$ = 152.1 °C, in good agreement with findings described by Fehri et al. [32]. With the introduction of plasticizer into polymeric matrix, the $T_g$ decreased (42.3 °C) suggesting the improvement in chain mobility. The PLA/ATBC sample showed the double melting temperatures ($T_m$) at 145.8 and 131.7 °C. It is noted that a shoulder is observed as endothermic melting points for all samples: PLA/AgNPs 0.5% (131.0 °C); PLA/AgNPs 1% (134.4 °C), and PLA/AgNPs 1.5% (131.0 °C),

meaning a bimodal distribution of crystallite size, different morphologies occurred in the heating and cooling process, and formation of disorder alpha phase [9].

From Table 3, it is also observed that the Col 5 was incorporated into the plasticized PLA an increase of $T_g$ was obtained compared to that for the plasticized PLA-based bionanocomposites. This increase is accentuated by the introduction of AgNPs. $X_c$ evaluated from the melting enthalpy of samples (using the Equation (1)) showed higher values for bionanocomposites compared with those of PLA/ATBC and PLA/Col 5 samples.

### 3.4. Chemiluminescence Analysis

The AgNPs concentration has a significant role in the delay of degradation in the polymer substrate. By comparing the control and PLA samples (Figure 4a,b), an accelerated oxidation occurred during isothermal CL measurements can be noticed at the highest testing temperature (200 °C). The oxidation period is much shorter in the presence of low content of silver nanoparticles. On the opposite side, at lower temperatures (180 °C and 190 °C) the emission intensities are significantly lower in the samples containing 0.5% AgNPs. At higher loadings of AgNPs (Figure 4c,d) the decrease in the CL intensities reveals the efficient inhibition of oxidation. This feature strengthens the idea that silver nanoparticles can be successfully used for the extension of material lifetimes. The amplitude of oxidation is diminished as the concentration of AgNPs increases. The spreading of these particles in the PLA matrix ensures the homogeneous stabilization of material, even though the diffusing oxygen would feed degradation.

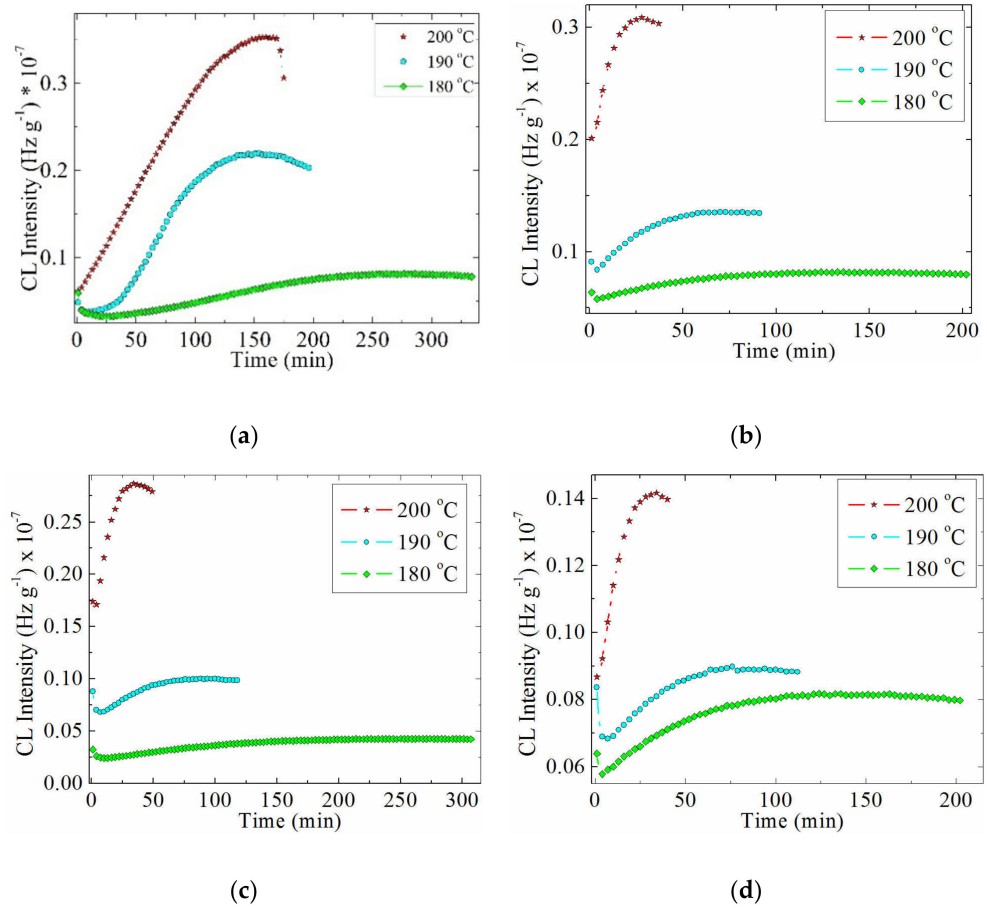

**Figure 4.** Isothermal runs of chemiluminescence intensity of PLA bionanocomposites: (**a**) PLA/ATBC; (**b**) PLA/AgNPs 0.5%; (**c**) PLA/AgNPs 1%; (**d**) PLA/AgNPs 1.5%, at three testing temperatures: 180 °C, 190 °C and 200 °C, Dose 0 kGy.

The stabilization effect of higher density of silver nanoparticles can be also obtained by light γ-irradiation. Figure 5 demonstrates the smoother oxidation of PLA when the amount of AgNPs is minimum 1%. The agglomeration of nanoparticles produces minimization of the distance between free radicals increasing the probability of their reactions with penetrating oxygen.

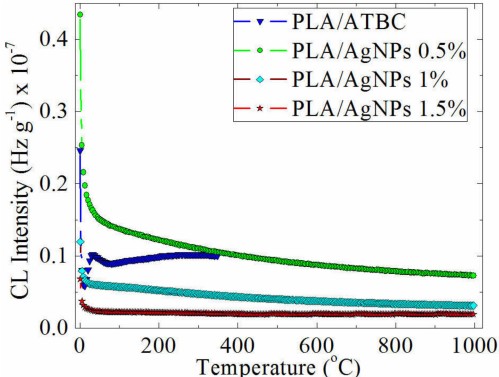

**Figure 5.** Nonisothermal chemiluminescence (CL) spectra recorded on irradiated PLA-based bionanocomposites at dose of 10 kGy.

The nonisothermal CL measurements (Figure 6a–c) are the proofs for the protective effect of AgNPs at different particle concentrations. The upper positions of curves depicting the oxidation of PLA loaded with 0.5% AgNPs prove the soft stabilization action at this lower concentration. The exposure to different low doses by γ-irradiation (10 and 20 kGy) does not alter the sequence of stabilization effects, but it may really assume that the slight degradation of polymer would be accompanied by an oxidation of silver nanoparticles. The increasing concentration of dispersed particles brings about similar level. However, the higher dose determines a faster oxidation of polymer phase on the region of superior temperatures. It means that the progress of oxidation aging of PLA is less hindered by modified silver nanoparticles when they are subjected to an intense energy transfer. For the explanation of the small difference between the contributions of inorganic component on the delay of oxidation the coalescence of nanoparticles takes place. The availability of these new particle consistence upon the delay of degradation is diminishes and the contribution of higher silver loading can be disregarded.

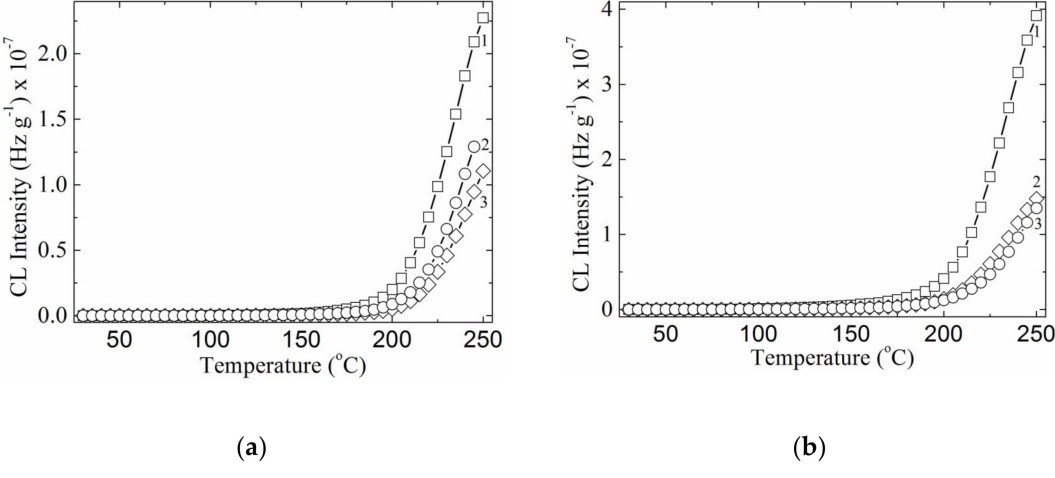

(**a**)　　　　　　　　　　　　　　(**b**)

**Figure 6.** *Cont.*

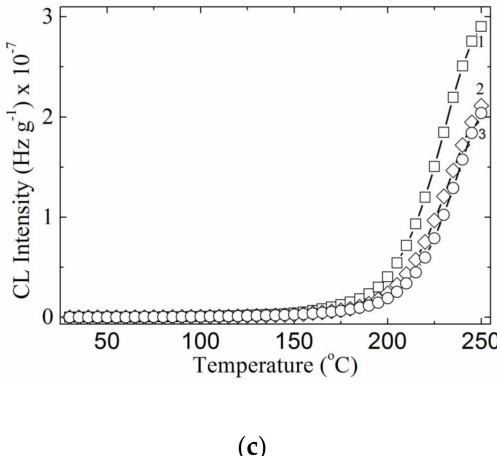

(**c**)

**Figure 6.** Chemiluminescence (CL) spectra registered on PLA-based samples: (**a**) 0 kGy; (**b**) 10 kGy; and (**c**) 20 kGy—nonisothermal measurements, heating rate: 5 °C min$^{-1}$. The notation 1, 2, and 3 signifies PLA/AgNPs 0.5%, PLA/AgNPs 1%, and PLA/AgNPs 1.5%, respectively.

### 3.5. In Vitro Cytotoxicity Evaluation

Cell viability assays were conducted in order to establish the cytotoxic potential of tested bionanocomposites films. The MTT results showed that all tested biocomposites had no cytotoxic activity (Figure 7a). The percentages of cell viability were higher than 80% for all biocomposites variants at both exposure times (24 and 48 h). PLA/ATBC and PLA/HC samples exhibited similar viability values at both 24 h and 48 h, which in turn were similar to the control (untreated cells) considered 100% viable. The PLA samples containing Ag nanoparticles showed values of cell viability higher than 100% after 24 h, but slightly decreased after 48 h (between 92.71% for PLA/AgNPs 0.5% and 96.43% for PLA/AgNPs 1.5%).

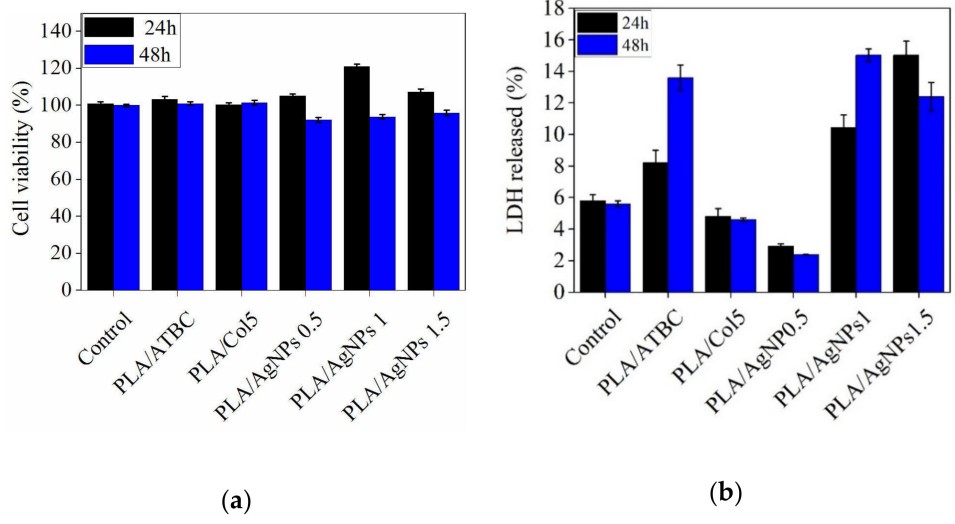

(**a**)　　　　　　　　　　　　　　　　　　　　　　(**b**)

**Figure 7.** Cell viability of NCTC clone L929 fibroblast cells cultivated in the presence of blends and bionanocomposites for 24 and 48 h, evaluated by 3-(4,5-dimethylthiazol-2-yl)-2,5-diphenyltetrazolium bromide (MTT) (**a**) and lactate dehydrogenase (LDH) (**b**) assays. The control was represented by untreated cells (cells cultivated in culture plate in Minimum Essential Medium (MEM). Data were presented as mean ± SD (*n* = 3).

Cells cultivated in the presence of the bionanocomposites were also investigated for their cell membrane integrity by measuring LDH activity in the culture medium. Our results showed that all tested samples had no cytotoxic effect on cells after 24 h and 48 h of cultivation and therefore, they have

not affected cell membrane integrity and cell viability (Figure 7b). PLA/AgNPs 0.5% sample showed the lowest percentages of LDH released in the culture medium at both exposure times (2.92% and 2.38% at 24 and 48 h, respectively), whereas PLA/Col presented values lower than 6% and similar to that of the control. PLA/AgNPs 1% and PLA/AgNPs 1.5% samples exhibited slightly higher percentages of the LDH released, ranging between 10.44% and 15.02%.

### 3.6. Cell Morphology

Light microscope images revealed a normal morphology of cells exposed to different bionanocomposites, similar to that of the control (Figure 8A–F).

Thus, cells had a normal fibroblastic phenotype, with euchromatic nuclei and 1–2 nucleoli, clear cytoplasm, and cytoplasmic extensions. Cell density was also similar to that of the untreated control, reaching an almost complete monolayer, with cells covering around 90%–95% of the well-surface.

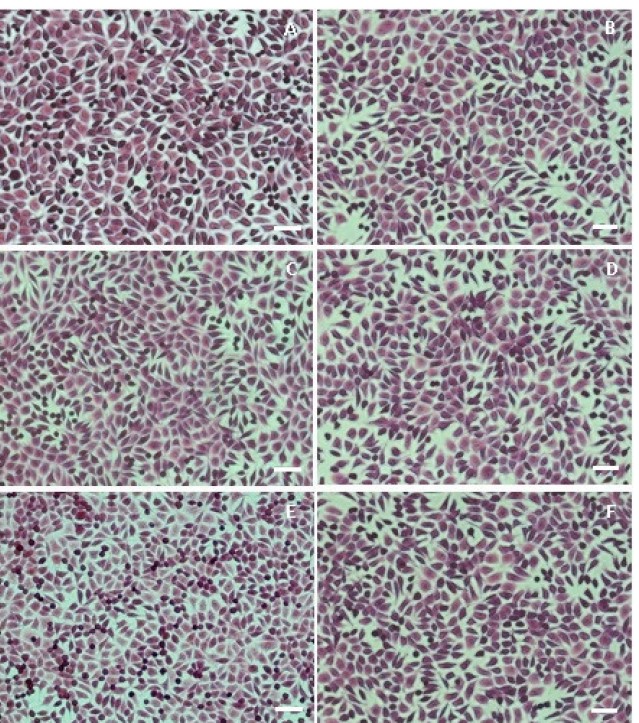

**Figure 8.** Light microscope images of mouse fibroblast cells untreated (control) (**A**) and treated with PLA/ATBC (**B**); PLA/Col (**C**); PLA/AgNPs 0.5% (**D**); PLA/AgNPs 1% (**E**), and PLA/AgNPs 1.5% (**F**) for 48 h (hematoxylin-eosin staining). Scale bar = 50 μm.

### 3.7. Effect of Bionanocomposites on Cell Cycle Distribution

In order to confirm the biocompatibility results of these new bionanocomposites, cell cycle distribution of cells treated with these samples was analyzed by flow cytometry. Changes observed in the cell cycle dynamics between the three distinct stages: G0/G1, S, and G2/M, were based on the measurement of the DNA content. Thus, the distribution of cells treated with different PLA bionanocomposites in the each phase of the cell cycle was similar to that of the control, indicating that all tested samples did not affect cell proliferation (Figure 9).

Cell treatment with bionanocomposite variants for 24 h induced an increase of cell population in the G0/G1 phase for all samples, ranging between 61.67% for PLA/Col and 66.39% for PLA/ATBC, compared to control sample (G1 53.9%). A decrease in the percentage of cells in S phase and little variation of the cell population in G2/M phase were also observed (Figure 10). Thus, the percentages of cell population in S phase ranged between 22.35% for PLA/AgNPs 1% and 28.12% for PLA/Col 5 and were slightly lower than the control (34.08%—Figure 10). The PLA/Ag 0.5% and PLA/Ag 1%

bionanocomposites have determined a slight increase of the cell population found in G2/M phase (13.74% and 13.95%, respectively), compared to the control culture (12.02%). In contrast, the PLA/ATBC, PLA/Col 5 and PLA/Ag 1.5% samples induced a slight decrease of the cell population in G2/M phase (Figure 10).

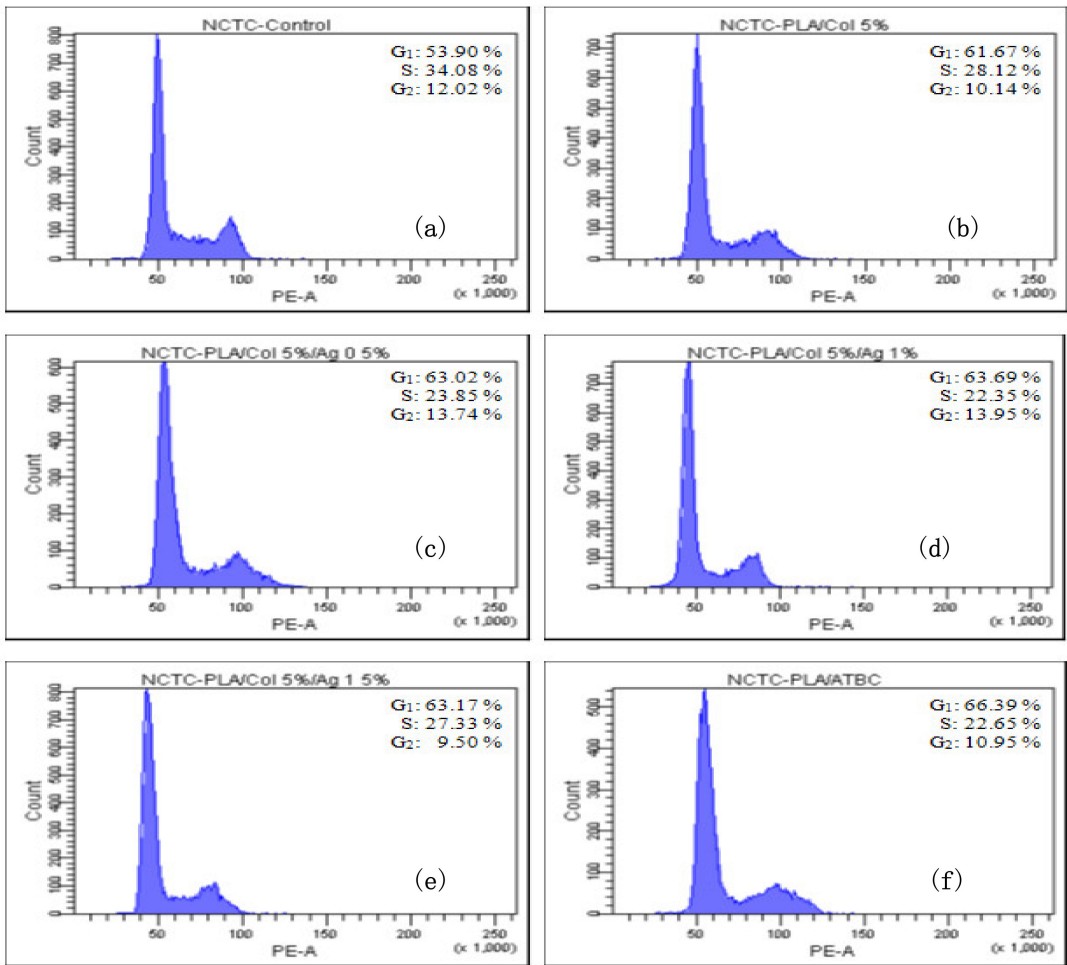

**Figure 9.** Cell cycle histograms of mouse fibroblast cells treated with different blends and bionanocomposites. (**a**) Control; (**b**) PLA/Col 5; (**c**) PLA/Col5; (**d**) PLA/AgNPs 1%; (**e**) PLA/AgNPs 1.5%; (**f**) PLA/ATBC.

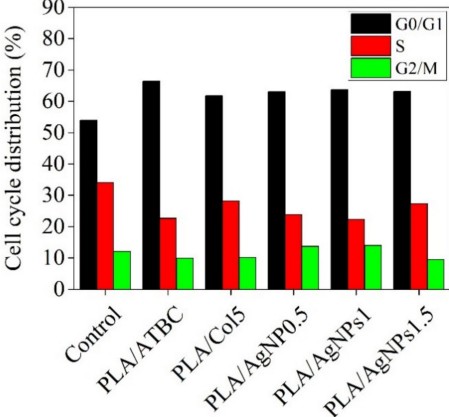

**Figure 10.** Percentage distribution in cell cycle phases of cells treated with PLA bionanocomposites.

## 4. Discussion

The use of natural biopolymers and nanofiller for biomedical coating applications has increased interest among researchers because of their biocompatibility and antimicrobial properties. In this study, PLA bionanocomposites containing plasticizer, hydrolyzed collagen and silver nanoparticles were obtained using melt mixing technology.

The topography of PLA bionanocomposites was evaluated by AFM. The results revealed the increased roughness in the case of PLA bionanocomposites due to the presence of collagen hydrolysate. The highest value of roughness surface was also observed in the case of reactive blending of PLA reinforced with thermoplastic cassava starch (TPCS) and commercial graphene (GRH) nanoplatelets [33]. The authors attributed the increase in roughness to irregular shapes of starch granules, while the presence of GRH nanoplatelets did not have a significant contribution.

FT-IR analysis revealed the presence of C–O, C–H, and O–H stretching of the –CH (CH$_3$)–OH end group of PLA [34]. The absorption peaks assigned to C=O stretch (1041–1180 m$^{-1}$) and –CH$_2$ asymmetrical stretch (2853–2923 cm$^{-1}$) demonstrated the existence of collagen into bionanocomposites [34]. The addition of silver nanoparticles into PLA bionanocomposites up to 1 wt.% led to increase in the intensity of main absorption bands for PLA/AgNPs 0.5 and PLA/AgNPs 1 bionanocomposites. The same effect was observed in the case of polylactide (PLA)/nano-TiO$_2$ and PLA/nano-TiO$_2$/nano-Ag blends films obtained by solvent casting [35]. This behavior can be explained by the presence of van der Waals interactions between the HO groups of PLA and the net atomic charge on the surface of the AgNPs [23] due to the asymmetric partition of electrons. The sample containing 1.5% AgNPs did not show any increase in the intensity of absorption bands, maybe due to the agglomeration of inorganic nanoparticles during melt processing.

DSC showed that the introduction of ATBC plasticizer into PLA led to the reduction in both T$_g$ and T$_m$, as a result of the increase of chain macromolecular mobility. This behavior was also reported in another paper [8]. Our results revealed an increase of the T$_g$ and the degree of crystallinity of PLA with loading of AgNPs indicated that a small amount of AgNPs acts as a nucleating agent for increase the crystallinity of the plasticized PLA [36]. Another paper stipulated that the introduction of AgNPs had no significant effect on the T$_g$ of PLA samples [30,35], while an increase in T$_g$ value was observed in the case of PLA/cassava/GRH nanoplatelets [33]. The increase in the X$_c$ was also reported in the case of PLA/chitosan foam [14].

The contribution of AgNPs to the durability of PLA bionanocomposites was investigated by chemiluminescence (CL). The stability of polylactic acid is related to the β-scissions occurred in the molecular backbones [37]. Study by chemiluminescence revealed that two types of oxidation precursors are implicated in the thermal degradation of PLA [38]. Accordingly, the addition of AgNPs will change the rate of degradation because they interact with free radicals. The contribution of AgNPs is connected to the electronic shell structure that is incomplete. The two contradictory processes, oxidation and stabilization, have an overall effect. It is observed that the silver nanoparticles block the oxidation of free radicals and the stability is improved. The same effect was observed in the case of introduction the natural phenolic antioxidants like powdered rosemary ethanolic extract into the PLA matrix [39]. The literature shows the evaluation of antioxidant efficacy of synthesized AgNPs using 1,1-diphenyl-2-picryl hydrazyl radical (DPPH) [40], H$_2$O$_2$ [41], ABTS radical cation scavenging, and FRAP assays [42]. It was reported that the DPPH and ABTS cation radical scavenging activity of AgNPs increased with the increase in the concentration of AgNPs [40,42]. The improved thermal stability of the antibacterial green nanocomposites containing PLA, poly(butylene adipate-co-terephthalate) (PBAT) and nanocrystal cellulose (NCC)-Ag nanohybrids was also demonstrated by thermal gravimetric analysis (TGA) [36]. It was explained by the higher decomposition activation energy correlated to the AgNPs incorporation [36].

The CL runs show that the intensity of the light emission is dependent on the tested temperatures and the amount of AgNPs into bionanocomposites. The maximum in the light emissions for all bionanocomposites at 180 and 190 °C is reached after 100 min, since at 200 °C it takes about 50

min. At the same time the intensity of the bionanocomposites light emissions at 200 °C decreased with the increase of antimicrobial agent. It should be noted that the hydroperoxide stability is in the order: AgNPs 1.5 > AgNPs 1 > AgNP 0.5. AgNPs could act as a nanocatalyst [43] on the PLA bionanocomposites. The results on the stability study appeared in PLA by the addition of AgNPs indicate their promising contribution to the durability of PLA based material.

In the present study, the cell viability of mouse fibroblast cells cultivated in the presence of PLA bionanocomposites was investigated by quantitative MTT and LDH assays. Cell morphology was also examined after 48 h of cell incubation in the presence of the samples using hematoxylin-eosin staining. Our results indicated a high degree of biocompatibility of all tested bionanocomposites, with viability values higher than 90% after 48 h of treatment. Furthermore, a good correlation between the cell morphological observations and the quantitative results obtained by MTT and LDH viability assays was also observed. Due to their good biocompatibility, the PLA-based materials were extensively used in the field of tissue engineering. However, in order to obtain better optimized scaffolds and to increase cell proliferation and cell adhesion on their surface, PLA have been mixed with other polymers, such as collagen, gelatin or graphene oxide [44,45]. For instance, Qiao et al. [45] found that PLA/collagen electrospun fibrous scaffolds at a 60:40 weight ratio showed greatest stability, cell attachment, cell proliferation, and osteogenic differentiation of bone marrow stromal cells after five-week culture period. Furthermore, hybrid fiber matrices composed of poly(d,ʟ-lactide-co-glycolide) (PLGA), collagen, and graphene oxide were found to exhibit superior biocompatibility as well as enhanced properties for the attachment and proliferation of the C2C12 skeletal myoblasts. These matrices also induced the spontaneous myogenic differentiation, results which recommend them as potential scaffolds for skeletal muscle regeneration [46]. Kang et al. [47] developed electrospun membranes based on biodegradable polylactic acid and hydrosoluble collagen for tissue engineering applications and the MTT assay used to evaluate the survival rate of L929 cells cultivated in the presence of the membranes indicated that they were suitable for the cell proliferation. In another study, poly(d,ʟ-lactide-co-glycolide) (PLGA) nanofibrous membranes embedded with collagen were also found to be stable and biocompatible, which made them suitable to promote tendon–bone interface integration [4]. Some in vitro studies have shown that silver nanoparticles can induce toxicity at different cell levels and can cause various health problems [46,48,49], but used in appropriate concentrations they can be good candidates for many applications in the medical field, mainly due to their antimicrobial and anti-inflammatory properties [50]. For instance, PLA-poly(ε-caprolactone) nanofibers containing lower amounts of AgNPs (0.25 wt.%) favored cell proliferation and maintained the normal cell morphology of human skin fibroblasts, suggesting them as suitable substrates for wound healing [51]. Liu et al. [52] fabricated polymer composites based on PLA, hydroxyapatite and AgNPs for bone tissue engineering applications. Based on in vitro tests, they found that high concentrations of AgNPs (18 wt.% and 25 wt.%) inhibited osteoblast proliferation, whereas lower levels (2 wt.% and 6 wt.%) induced good cytocompatibility.

The cell cycle distribution oh mouse fibroblast NCTC, clone 929, cultivated in the presence of tested PLA-based bionanocomposites was also investigated by flow cytometry. Overall, the dynamics of the cell were not blocked in any phase of the cell cycle and were prepared for a new cell division. This pattern observed in the dynamics of the cell cycle is in agreement with the cell cycle distribution of cells at an optimal density. G1 phase is very important because it influences the cell to divide or not, whereas in G2 phase cells grow in size and synthesize mRNA and proteins required for DNA synthesis.

Silver nanoparticles were reported to be toxic to human cells at concentrations above 1% [53] and induce the reactive oxygen species generation [54]. It has been reported that the cytotoxicity of antimicrobial medical devices containing silver nanoparticles is dependent on the size and surface area of inorganic ions released from the devices [55]. Contradictory data showed that the AgNPs with a mean size of 15 nm induced the greatest loss in mitochondrial activity [56]. However, according to the Occupational Safety and Health Administration (OSHA) and the Mine Safety and Health Administration (MSHA), the permissible exposure limit for silver compounds are estimated to be 0.01 mg/m$^3$ [57]. The evidence related the commercial antimicrobial Luer-activated connector coated with

silver nanoparticles, showed it is approved by the Food and Drug Administration [58]. However, the low concentration of silver nanoparticles is needed to ensure both non-cytotoxicity and bactericidal effect of elaborated materials. The values obtained in this experiment indicated that the cells were not blocked in any phase of the cell cycle and that most of these cells proliferated. All tests recommended the variant containing 0.5% AgNPs for biomedical applications. Future research will be conducted to investigate the release pattern of AgNPs in the medium, as well as their interaction with the cells.

The effect of the silver nanoparticles on the antimicrobial activity of the prepared PLA-based bionanocomposites will be evaluated as the future perspectives of these materials to use as biomedical coating.

## 5. Conclusions

In this, study, the physical, chemical, and biological properties of bionanocomposites containing PLA, hydrolyzed collagen (5%), and 0.5%–1.5% AgNPs have been studied using spectral characteristics (ATR FT-IR), surface structure (AFM), thermal properties (DSC), and thermal stability by chemiluminescence (CL), as well as in vitro biocompatibility. The results revealed that by introducing of AgNPs into the plasticized PLA, the surface became rough and the crystallinity and glass transition increased, as confirmed by the AFM and DSC analyses. The collagen and AgNPs led to the increase both of $T_g$ and $X_c$ for the plastified PLA bionanocomposites. The addition of AgNPs showed a significant effect in the block of the oxidation of PLA free radicals, thus the stability of bionanocomposites was improved. The cell viability of mouse fibroblast cell line on PLA bionanocomposites investigated on NCTC, clone L929, using quantitative MTT and LDH activity assays revealed a high degree of biocompatibility for all tested PLA-based bionanocomposites. It was found that an optimum content of 0.5% antimicrobial agent assured a roughness surface of 46.5 nm, a soft stabilization action at 10 kGy, good cell viability of NCTC clone L929 fibroblast cells and the lowest percentages of LDH released in the culture medium at 24 and 48 h exposure being recommended for the development of biomedical coating materials The tested materials will be further explored to evaluate the effect of AgNPs on the antimicrobial properties of bionanocomposites.

**Author Contributions:** Conceptualization—M.R., I.A., and E.M.; Formal analysis—I.A., C.P., and E.M.; Methodology—M.R., I.A., and E.M.; Investigation (melt processing and FT-IR)—M.R.; Investigation (biocompatibility)—L.M.S.; Investigation (chemiluminescence)—T.Z.; Investigation (flow cytometry)—A.M.S.; Investigation (thermal analysis)—A.A.Ţ.; Investigation (AFM)—A.M.P.; Writing–original draft—M.R.; Validation—I.A., E.M., and C.P. All authors have read and agreed to the published version of the manuscript.

**Funding:** This research was funded by the POC Program, Grant no. 49/05.09.2016, Project ID P_37_649 and the grant of the Romanian National Authority for Scientific Research and Innovation within Programme Nucleu, project no 25N /2019 (19270102).

**Conflicts of Interest:** The authors declare no conflict of interest.

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
