# Peer review of "Development of Bionanocomposites Based on PLA, Collagen and AgNPs and Characterization of Their Stability and In Vitro Biocompatibility"

_applsci, doi:10.3390/app10072265_

Round 1

Reviewer 1 Report

Journal: Applied Sciences

Title: In vitro Biocompatibility and Stabilization Effect of AgNPs on Some Plasticized PLA/Collagen Bionanocomposites

Authors: Maria Râpă, Laura Mihaela Stefan, Traian Zaharescu, Ana-Maria Seciu, Anca Andreea Èšurcanu, Ecaterina Matei, Andra Mihaela Predescu, Iulian Antoniac, Cristian Predescu

OVERALL COMMENTS/DECISION:

The paper studies the development of bionanocomposites based on PLA, collagen and AgNPs using the melt mixing technique. The developed bionanocomposites in the form of films are characterized regarding their thermal, optical and chemical stability properties. Furthermore, in vitro experiments with fibroblast cell line have been performed in order to evaluate the cytotoxicity and biocompatibility of the films. The study reports that the addition of the AgNPs increased the Tg and contributes to the improved stability of the films. All films supported the growth of the fibroblast cells for 48h without compromising the viability.  

The work presented in this paper is a continuation of the work of the research group on the development and characterization of polymer-nanoparticles bionanocomposites [Rapa, M. et al. PLA/collagen hydrolysate/silver nanoparticles bionanocomposites for potential antimicrobial urinary drains. Polym-Plast Technol 2019; Rapa, M. et al. Effect of hydrolyzed collagen on thermal, mechanical and biological properties of poly(lactic acid) bionanocomposites. Iran Polym J 2019, 28] and as a whole exhibits a good scientific background.

The Introduction section needs restructuring and the problem statement of the study should be more precisely explained (please see Major Comment #5). The Materials and Methods section includes most of the necessary experimental information of the study, except for the case of the in vitro experiments (please see Major Comment #7).  The Results are generally presented in a clear manner, but the section would benefit from restructuring (please see Major Comment #8).  The correlation of the results with the respective literature, as presented in the Discussion section, is good regarding the material characterization part but could be enriched in the case of the in vitro study. Furthermore, the Discussion section should be enriched with the future perspectives of the developed bionanocomposite. The targeted application of the developed bionanocomposites is not clear in the manuscript in its present form.  

Major Comments

  1. Title: The title could be more precise and informative on the novelty. A suggestion could be ‘Development of bionanocomposites based on PLA, collagen and AgNPs and characterization of their stability and in vitro biocompatibility’.
  2. Abstract-1: In the Abstract it is mentioned that ‘Antimicrobial bionanocomposites including PLA, ECM and AgNPS were prepared…’. Given that no study of their antimicrobial properties was performed, the term antimicrobial is misleading. Testing of their antimicrobial properties can be described in the future perspectives of the study the Discussion
  3. Abstract-2: In the Abstract it is mentioned that ‘The obtained results recommend the potential use of PLA bionanocomposites for tissue engineering’. Just by showing the biocompatibility of the developed bionanocomposites does not suffice for this claim. The authors could suggest the biomedical coatings as a future perspective.
  4. Abstract-3: In the Abstract it is mentioned that scaffolds have been prepared. The term scaffold is not used from the authors in any other point of the text. It would be more precise to use the term ‘films’ instead.
  5. Introduction: The Introduction section is organized into too many (i.e. 8) small paragraphs. A restructuring and merge of the paragraphs into fewer but consistent ones are highly suggested. A suggestion regarding the structure is the following:
  • Paragraph: ’PLA’
  • Paragraph: PLA/Collagen composites (with an emphasis on biocompatibility and antimicrobial properties)
  • Paragraph: ‘AgNPs for their antimicrobial properties’
  • Paragraph: ‘AgNPs and polymers composites’
  • Paragraph: ‘The aim of the present study’
  1. Introduction, Line 56: ‘and low osteogenic differentiation’. Given that the present study is not focusing on the osteogenic differentiation of cells, this detail is irrelevant and can be omitted.
  2. Materials & Methods (M&Ms): It is not clear in which form the PLA/collagen composites have been used in the in vitro experiments with cells. Are they in the form of films? If yes, this information shall be defined in the M&Ms section. For example, in the Paragraph 2.3.6 (Lines 165: ‘The bionanocomposite formulations were added into the wells’ this is not clear at all. The way it is written is as if the composites are added on top of the cells. Furthermore, are the films sterilized? Are the films pre-conditioned in the growth medium before the addition of cells?
  3. Results: There are too many Figures explaining the results. Some Figures could be merged into one, given that they refer to the same characterization technique. As a suggestion:
  • Figures 5-7 refer to the chemiluminescence data and should be merged into a single Figure
  • Figures 8 and 9 refer to the in vitro cytotoxicity of the composites and should be merged into a single Figure
  1. Results- In vitro study: It is not clear how many times have the in vitro experiments been repeated. In the M&M section, it is stated that ‘all samples were tested in triplicates’. Does this mean triplicates per experiment? The results of the in vitro experiments results cannot suffice without experimental repeats (the experiment must have been performed at least 3 individual times).
  2. Results-In vitro study/LDH activity assay-1: With which equation have the percentages of LDH absorbance been calculated? How can the results in the evaluation of cytoxicity (Figure 9) be compared with the LDH positive control of the assay? Have the authors measured the absorbance when using the LDH positive control?
  3. Results-In vitro study/LDH activity assay-2: How can the authors explain the huge increase in LDH with increasing the concentration of AgNPs (i.e. by comparing the values for PLA/AgNP0.5 and PLA/AgNPs)? How many times has this experiment been repeated?
  4. Results-In vitro study/Cell morphology: High (instead of low) magnification images are required for cell morphology evaluation.
  5. Discussion: The paragraph discussing the FTIR data (Lines 396-397) needs to be enriched with respective literature.
  6. Conclusion: This should be rewritten as a single paragraph and has to include one sentence with the future perspective of the developed materials or experiments, e.g. the study of the antimicrobial properties.

Minor Comments

In the following section, there are some points that must be corrected or rephrased.

  1. Page 1, Line 20: Please correct ‘extracellular matrix’ into ‘collagen’.
  2. Page 1, Line 21: Please correct ‘Ag nanoparticles’ into ‘silver nanoparticles’.
  3. Page 1, Line 22: Please delete the sentence ‘were evaluated by DSC measurements’ (given that the other characterization techniques are not mentioned in the Abstract).
  4. Page 3, Line 119: Please include title ‘2.3.1 UV-Vis spectroscopy’.
  5. Page 4, Line 167: Please correct ‘the cells’ into ‘cells’.
  6. Page 4, Lines 214-215: Please correct ‘it is observed the broader peak (Figure 1)’ into ‘a broader peak in the wavelength range of VALUE is observed’.
  7. Page 7, Line 231: Please correct ‘see-island’ into ‘sea-island’.
  8. Page 8, Line 259: What are the numbers ‘152.1; 59.9’ ?
  9. Page 8, Line 275: Please correct ‘If the control and PLA samples are compared’ into ‘By comparing the control and PLA samples,’.
  10. Page 8, Line 283: Please correct ‘higher and higher’ into ‘increases’.
  11. Page 10, Line 307: Please correct ‘Thus, the MTT’ into ‘The MTT’.
  12. Figure 10: Please add a scale bar
  13. Figure 10: Please use different color for the captions. The captions should better be positioned on the top left or right of the image.
  14. Page 13, Lines 398-399: The sentence ‘DSC curves…showed’ is suggested to be written ‘DSC showed’.
  15. Page 13, Line 391: Please correct ‘the higher’ into ‘the highest’.
  16. Page 14, Line 427: Please correct ‘the cell viability of PLA bionanocmposites was investigated on mouse fibroblast cell line’ into ‘the cell viability of mouse fibroblast cell line on PLA bionanocmposites was investigated’
  17. Page 14, Line 446: Please correct ‘because influences’ into ‘because it influences’.
  18. Page 14, Line 447: Please correct ‘whereas in G1’ into ‘whereas in G2’.
  19. Page 14, Line 451: Please correct ‘Conclusions’ into ‘Conclusion’.
  20. Page 14, Line 452: Please correct ‘The investigation on the physical’ into ‘In this, study, the physical’.
  21. Page 14, Line 453: Please correct ‘is performed by the study of’ into ‘have been studied using’.
  22. Page 14, Line 457: Please correct ‘The result obtained’ into ‘The results’.
  23. Line 513: In Reference #14 the name of the journal is in bold.
  24. Line 574: The title of Reference #38 is missing.

Author Response

We would like to sincerely thank you and the referees for evaluating our manuscript entitled “In vitro Biocompatibility and Stabilization Effect of AgNPs on Some Plasticized PLA/Collagen Bionanocomposites”. We found the comments and suggestions of the reviewers very constructive and we accordingly revised the manuscript. All changes are highlighted in the revised manuscript. Please find attached below this letter, the point-to point answers to the referees.

Reviewer 1

The Authors of the manuscript entitled In vitro Biocompatibility and Stabilization Effect of AgNPs on Some Plasticized PLA/Collagen Bionanocomposites, submitted to Applied Sciences thank the referee for reviewing our manuscript. We are deeply grateful for the advice and for the observations and comments which we addressed and feel that greatly increased the quality of our manuscript. Please find below the answers to all questions and suggestions.

The paper studies the development of bionanocomposites based on PLA, collagen and AgNPs using the melt mixing technique. The developed bionanocomposites in the form of films are characterized regarding their thermal, optical and chemical stability properties. Furthermore, in vitro experiments with fibroblast cell line have been performed in order to evaluate the cytotoxicity and biocompatibility of the films. The study reports that the addition of the AgNPs increased the Tg and contributes to the improved stability of the films. All films supported the growth of the fibroblast cells for 48h without compromising the viability.  

The work presented in this paper is a continuation of the work of the research group on the development and characterization of polymer-nanoparticles bionanocomposites [Rapa, M. et al. PLA/collagen hydrolysate/silver nanoparticles bionanocomposites for potential antimicrobial urinary drains. Polym-Plast Technol 2019; Rapa, M. et al. Effect of hydrolyzed collagen on thermal, mechanical and biological properties of poly(lactic acid) bionanocomposites. Iran Polym J 2019, 28] and as a whole exhibits a good scientific background.

The Introduction section needs restructuring and the problem statement of the study should be more precisely explained (please see Major Comment #5). The Materials and Methods section includes most of the necessary experimental information of the study, except for the case of the in vitro experiments (please see Major Comment #7).  The Results are generally presented in a clear manner, but the section would benefit from restructuring (please see Major Comment #8).  The correlation of the results with the respective literature, as presented in the Discussion section, is good regarding the material characterization part but could be enriched in the case of the in vitro study. Furthermore, the Discussion section should be enriched with the future perspectives of the developed bionanocomposite. The targeted application of the developed bionanocomposites is not clear in the manuscript in its present form.  

Major Comments

  1. Title: The title could be more precise and informative on the novelty. A suggestion could be ‘Development of bionanocomposites based on PLA, collagen and AgNPs and characterization of their stability and in vitro biocompatibility’.

Answer: Thank you for your suggestion which is very useful. The title was changed accordingly.

  1. Abstract-1: In the Abstract it is mentioned that ‘Antimicrobial bionanocomposites including PLA, ECM and AgNPS were prepared…’. Given that no study of their antimicrobial properties was performed, the term antimicrobial is misleading. Testing of their antimicrobial properties can be described in the future perspectives of the study the Discussion

Answer: In this form, the Abstract does not contain “the antimicrobial” word. Discussion section was completely with the evaluation of antimicrobial activity of the prepared bionanocomposites, as future perspective of this study.

  1. Abstract-2: In the Abstract it is mentioned that ‘The obtained results recommend the potential use of PLA bionanocomposites for tissue engineering’. Just by showing the biocompatibility of the developed bionanocomposites does not suffice for this claim. The authors could suggest the biomedical coatings as a future perspective.

Answer: The biomedical coatings of the prepared bionanocomposites replace the tissue engineering applications.

  1. Abstract-3: In the Abstract it is mentioned that scaffolds have been prepared. The term scaffold is not used from the authors in any other point of the text. It would be more precise to use the term ‘films’ instead.

Answer: The term “films” was put instead of “scaffold”.

  1. Introduction: The Introduction section is organized into too many (i.e. 8) small paragraphs. A restructuring and merge of the paragraphs into fewer but consistent ones are highly suggested. A suggestion regarding the structure is the following:
  • Paragraph: ’PLA’
  • Paragraph: PLA/Collagen composites (with an emphasis on biocompatibility and antimicrobial properties)
  • Paragraph: ‘AgNPs for their antimicrobial properties’
  • Paragraph: ‘AgNPs and polymers composites’
  • Paragraph: ‘The aim of the present study’

Answer: Thank you for your suggestion. The Introduction was organized into five paragraphs.

  1. Introduction, Line 56: ‘and low osteogenic differentiation’. Given that the present study is not focusing on the osteogenic differentiation of cells, this detail is irrelevant and can be omitted.

Answer: “low osteogenic differentiation” was omitted from this form of manuscript.

  1. Materials & Methods (M&Ms): It is not clear in which form the PLA/collagen composites have been used in the in vitro experiments with cells. Are they in the form of films? If yes, this information shall be defined in the M&Ms section. For example, in the Paragraph 2.3.6 (Lines 165: ‘The bionanocomposite formulations were added into the wells’ this is not clear at all. The way it is written is as if the composites are added on top of the cells. Furthermore, are the films sterilized? Are the films pre-conditioned in the growth medium before the addition of cells?

Answer: In the in vitro tests, PLA/collagen composites were used as films. All variants were cut in 5x5 mm samples and sterilized under UV light for 4 h. We used the direct contact method according to the SR EN ISO 10993-5:2009 standard, which implies adding the samples directly in the culture medium, on top of the cells, after 24 h from cell seeding. We modified the text in the Material and Methods section in order to clarify all the above aspects.

  1. Results: There are too many Figures explaining the results. Some Figures could be merged into one, given that they refer to the same characterization technique. As a suggestion:
  • Figures 5-7 refer to the chemiluminescence data and should be merged into a single Figure
  • Figures 8 and 9 refer to the in vitro cytotoxicity of the composites and should be merged into a single Figure

Answer: Your proposition regarding the overlapping of figs. 5-7 is not applicable, because an unreadable figure would result or the inclusion of these figures into one representation does not match, because they depict different results they are associated each other only for the explanation of sample behavior. The provided information will be unexplainably mixed and the reader’s understanding remains unclear. Please, look on other our published papers:

  1. Zaharescu, I. Blanco, F. A. Bottino. Surface antioxidant activity of modified particles in POSS/EPDM hybrids. Appl. Surf. Sci., 509, 144702 (2020). Doi:10.1016/j. apsusc.2019.144702.
  2. Zaharescu, C. Tardei, V. Marinescu, M. Râpă, M. Iordoc. Interphase surface effects on the thermal stability of hydroxyapatite/poly(lactic acid) hybrids. Ceramics Int., Doi: 10.1016/j.ceramint.2019.11.223.

The presentation of chemiluminescence results is similarly done.

Regarding the Figures 8 and 9, we merged them into one single figure, Figure 7 (a) (cell viability by MTT) and (b) (cell viability by LDH).

  1. Results- In vitro study: It is not clear how many times have the in vitro experiments been repeated. In the M&M section, it is stated that ‘all samples were tested in triplicates’. Does this mean triplicates per experiment? The results of the in vitro experiments results cannot suffice without experimental repeats (the experiment must have been performed at least 3 individual times).

Answer: All samples were tested in triplicate per experiment, and the experiments were performed 3 times. We modified the text in the Material and Methods section, please see the Section 2.3.5. In Vitro Cytotoxicity Tests.

  1. Results-In vitro study/LDH activity assay-1: With which equation have the percentages of LDH absorbance been calculated? How can the results in the evaluation of cytoxicity (Figure 9) be compared with the LDH positive control of the assay? Have the authors measured the absorbance when using the LDH positive control?

Answer: The results of the LDH assay were calculated using the following equation:

                         LDH released (%) = LDH medium (OD490) / LDH positive control (OD490) x 100                 (2);

The results were compared with the LDH positive control considered 100 % cytotoxic, all samples showing low cytotoxicity percentages (below 20 %), and correlated with those obtained by the MTT assay.

  1. Results-In vitro study/LDH activity assay-2: How can the authors explain the huge increase in LDH with increasing the concentration of AgNPs (i.e. by comparing the values for PLA/AgNP0.5 and PLA/AgNPs)? How many times has this experiment been repeated?

Answer: Although an increase in the percentage of LDH release could be observed with increasing the concentration of AgNPs, this increase is lower than the threshold of 20 %, which indicate a non-cytotoxic effect. Most probably, more AgNPs are released in the culture medium with the increasing concentration of AgNPs and this could affect cell integrity, but these alterations seem to be minor and in the limits of a good cytocompatibility. All samples were tested in triplicate per experiment, and the experiments were performed 3 times.

  1. Results-In vitro study/Cell morphology: High (instead of low) magnification images are required for cell morphology evaluation.

Answer: Unfortunately, we cannot provide high magnification images, but we consider that the provided images allow a good evaluation of cell morphology and also an estimation of the degree of cell proliferation and density.

  1. Discussion: The paragraph discussing the FTIR data (Lines 396-397) needs to be enriched with respective literature.

Answer: The paragraph discussing the FTIR data was enriched with respective literature [23,34,35].

  1. Conclusion: This should be rewritten as a single paragraph and has to include one sentence with the future perspective of the developed materials or experiments, e.g. the study of the antimicrobial properties.

Answer: The Conclusion was organized into a single paragraph containing the future perspective of the developed materials.

Minor Comments

In the following section, there are some points that must be corrected or rephrased.

  1. Page 1, Line 20: Please correct ‘extracellular matrix’ into ‘collagen’.
  2. Page 1, Line 21: Please correct ‘Ag nanoparticles’ into ‘silver nanoparticles’.
  3. Page 1, Line 22: Please delete the sentence ‘were evaluated by DSC measurements’ (given that the other characterization techniques are not mentioned in the Abstract).
  4. Page 3, Line 119: Please include title ‘2.3.1 UV-Vis spectroscopy’.
  5. Page 4, Line 167: Please correct ‘the cells’ into ‘cells’.
  6. Page 4, Lines 214-215: Please correct ‘it is observed the broader peak (Figure 1)’ into ‘a broader peak in the wavelength range of VALUE is observed’.
  7. Page 7, Line 231: Please correct ‘see-island’ into ‘sea-island’.
  8. Page 8, Line 259: What are the numbers ‘152.1; 59.9’ ?
  9. Page 8, Line 275: Please correct ‘If the control and PLA samples are compared’ into ‘By comparing the control and PLA samples,’.
  10. Page 8, Line 283: Please correct ‘higher and higher’ into ‘increases’.
  11. Page 10, Line 307: Please correct ‘Thus, the MTT’ into ‘The MTT’.
  12. Figure 10: Please add a scale bar
  13. Figure 10: Please use different color for the captions. The captions should better be positioned on the top left or right of the image.
  14. Page 13, Lines 398-399: The sentence ‘DSC curves…showed’ is suggested to be written ‘DSC showed’.
  15. Page 13, Line 391: Please correct ‘the higher’ into ‘the highest’.
  16. Page 14, Line 427: Please correct ‘the cell viability of PLA bionanocomposites was investigated on mouse fibroblast cell line’ into ‘the cell viability of mouse fibroblast cell line on PLA bionanocmposites was investigated’
  17. Page 14, Line 446: Please correct ‘because influences’ into ‘because it influences’.
  18. Page 14, Line 447: Please correct ‘whereas in G1’ into ‘whereas in G2’.
  19. Page 14, Line 451: Please correct ‘Conclusions’ into ‘Conclusion’.
  20. Page 14, Line 452: Please correct ‘The investigation on the physical’ into ‘In this, study, the physical’.
  21. Page 14, Line 453: Please correct ‘is performed by the study of’ into ‘have been studied using’.
  22. Page 14, Line 457: Please correct ‘The result obtained’ into ‘The results’.
  23. Line 513: In Reference #14 the name of the journal is in bold.
  24. Line 574: The title of Reference #38 is missing.

Answer: Thank you for all minor comments. All indicated points were corrected or rephrased, as you kindly suggested.

Reviewer 2 Report

There are some minor comments which can be replied before the acceptance

  1. In Fig. 8 there is no control with which the cytotoxicity of the material was compared.
  2. It is known: as a result of the hydrolysis of polylactide, lactic acid is released that leads to the acidification of the environment and  reduces the cell mass increase. Fibroblasts are especially sensitive to this negative effect. How are the authors going to prevent acidification from polylactide? The pH values of materials with silver nanoparticles and polylactide should be given.
    3. The investigated material is bioresorbable. What is happening with the silver nanoparticles after resorption?

Author Response

We would like to sincerely thank you and the referees for evaluating our manuscript entitled “In vitro Biocompatibility and Stabilization Effect of AgNPs on Some Plasticized PLA/Collagen Bionanocomposites”. We found the comments and suggestions of the reviewers very constructive and we accordingly revised the manuscript. All changes are highlighted in the revised manuscript. Please find attached below this letter, the point-to point answers to the referees.

The Authors of the manuscript entitled In vitro Biocompatibility and Stabilization Effect of AgNPs on Some Plasticized PLA/Collagen Bionanocomposites, submitted to Applied Sciences thank the referee for reviewing our manuscript. We are deeply grateful for the advice and for the observations and comments which we addressed and feel that greatly increased the quality of our manuscript. Please find below the answers to all questions and suggestions.

There are some minor comments which can be replied before the acceptance.

  1. In Fig. 8 there is no control with which the cytotoxicity of the material was compared.

Answer: Thank you for your comment and useful advice. The control cytotoxicity was added into the Figure 7 (a) as columns.

  1. It is known as a result of the hydrolysis of polylactide, lactic acid is released that leads to the acidification of the environment and reduces the cell mass increase. Fibroblasts are especially sensitive to this negative effect. How are the authors going to prevent acidification from polylactide? The pH values of materials with silver nanoparticles and polylactide should be given.

Answer: In vitro experiments performed for 48 hours, the pH of materials was not checked; it was assumed to be maintained at neutral pH value. According to other paper, the strategy to prevent the acidification due to PLA is to incorporate basic salts (calcium carbonate, sodium bicarbonate) [C. Mauli Agrawal, Kyriacos A. Athanasiou. Technique to Control pH in Vicinity of Biodegrading PLA-PGA Implants. Journal of Biomedical Materials Research, 38(2), 105–114. doi:10.1002/(sici)1097-4636(199722)].

  1. The investigated material is bioresorbable. What is happening with the silver nanoparticles after resorption?

Answer: Indeed, silver nanoparticles were found out to be toxic to human cells at concentrations above 1% [54] and induce the reactive oxygen species production [55]. It has been reported that the cytotoxicity antimicrobial medical devices containing silver nanoparticles is dependent on the size and surface-area of inorganic ions released from the devices [56]. Contradictory data showed that the AgNPs with a mean size of 15 nm induced the greatest loss in mitochondrial activity [57].

However, according to The Occupational Safety and Health Administration (OSHA) and the Mine Safety and Health Administration (MSHA), the permissible exposure limit for silver compounds are estimated to be of 0,01 mg/m3 [58]. The evidence related the commercial antimicrobial Luer-activated connector coated with silver nanoparticles, was approved by the Food and Drug Administration [59].

Discussion section was modified accordingly.

Reviewer 3 Report

The flow cytometry charts should be added or supplied

the quality of some figures need to be improved to be more clear

In Figure 9. Percentages of cytoplasmic LDH released by NCTC clone L929 cells treated, what is the explanation of the decrease in % of LDH released at 48 hr than 24 hr in case of PLA/AgNPs 0.5 & 1.5%.

The conclusion needs some revision to be more precise, for example; An optimum content of 0.5% antimicrobial agent

Author Response

We would like to sincerely thank you and the referees for evaluating our manuscript entitled “In vitro Biocompatibility and Stabilization Effect of AgNPs on Some Plasticized PLA/Collagen Bionanocomposites”. We found the comments and suggestions of the reviewers very constructive and we accordingly revised the manuscript. All changes are highlighted in the revised manuscript. Please find attached below this letter, the point-to point answers to the referees.

The Authors of the manuscript entitled In vitro Biocompatibility and Stabilization Effect of AgNPs on Some Plasticized PLA/Collagen Bionanocomposites, submitted to Applied Sciences thank the referee for reviewing our manuscript. We are deeply grateful for the advice and for the observations and comments which we addressed and feel that greatly increased the quality of our manuscript. Please find below the answers to all questions and suggestions.

  1. The flow cytometry charts should be added or supplied

Answer: Figure 9 showed the cell cycle histograms of mouse fibroblast cells treated with different blends and bionanocomposites was added.

  1. the quality of some figures need to be improved to be more clear

Answer: The Figure 2 (FTIR), Figure 3 (DSC), Figure 6 (nonisothermal CL measurements), and Figure 8 (cell morphology) were presented more clear.

  1. In Figure 9. Percentages of cytoplasmic LDH released by NCTC clone L929 cells treated, what is the explanation of the decrease in % of LDH released at 48 hr than 24 hr in case of PLA/AgNPs 0.5 & 1.5%.

Answer: In the first 24 hours, a higher release of the AgNPs in the culture medium may occur and could affect cell integrity. However, the differences observed in the percentages of the LDH released between 24 hours and 48 hours are quite low (0.54% for PLA/AgNPs 0.5% and 2.67% for PLA/AgNPs 1.5%).

In addition, the in vitro biocompatible profile determined in this paper represents preliminary data in order to evaluate the potential of these bionanocomposites to be used in biomedical applications. Based on these results, future research will be conducted to investigate the release pattern of AgNPs in the medium, as well as their interaction with the cells.

  1. The conclusion needs some revision to be more precise, for example; An optimum content of 0.5% antimicrobial agent

Answer: The collagen and AgNPs had a synergetic effect leading to the increase both of Tg and Xc for the plastified PLA bionanocomposites. The addition of AgNPs showed a significant effect in the block of the oxidation of PLA free radicals, thus the stability of bionanocomposites was improved. The cell viability of mouse fibroblast cell line on PLA bionanocomposites investigated on NCTC, clone L929, using quantitative MTT and LDH activity assays revealed a high degree of biocompatibility for all tested PLA-based bionanocomposites. It was found that an optimum content of 0.5% antimicrobial agent assured a roughness surface of 46.5 nm, a soft stabilization action at 10 kGy, good cell viability of NCTC clone L929 fibroblast cells and the lowest percentages of LDH released in the culture medium at 24 h and 48 h exposure being recommended for the development of biomedical coating materials The tested materials will be further explored to evaluate the effect of AgNPs on the antimicrobial properties of bionanocomposites.

The Conclusion was revised accordingly.

Reviewer 4 Report

The paper present an extended full characterization of the subject antimicrobial bio-nanocomposites . It is an useful report but with some not correct evaluation and poor scientific speculation. Some week points needing revision are indicated

1).The objective of the work, last sense of the introduction , is not a motivation but rather a synthetic abstract !Should be implemented indicating the possible importance of the study for knowledge and application.

2).UV spectra provide no information . The section could be removed

3).Normalized FT-IT^R spectra are of poor quality and not adequately resolved to allow determining      frequency of absorption

4).It is impossible to me the understanding of how values reported in Table 3 were evaluated from Figure 4 showing DSC traces. In this last there is no evidence of Tg .Also the type of crystallinity of the sample with 1% Agnps appears much different from all other samples which look similar .WHY ?

5).The comments from line 295 to 301 and the fig 7 a)-c) again reports some not clear and probably not correctly corresponding data.

6) with ref to 5) the lines 296-7 report “increasing irradiation dose causes the progress in the process of oxidation happened in PLA/AgNPs 0.5%,”. This is not what fig 7 a indicates

7). Also the lines 298-9 say “.. slight irradiation at 10 kGy brings about a diminution of oxidation levels in comparison with the exposure at superior dose of 20 kGy”. Why and how ???Hard to accept.

8)The discussion is merely descriptive of the results without critics and comparison with supporting literature. For instance the free radical inhibition of Agnps is claimed without confirming support.

9) the concluding section is irrelevant , just a synthetic summary.

Author Response

We would like to sincerely thank you and the referees for evaluating our manuscript entitled “In vitro Biocompatibility and Stabilization Effect of AgNPs on Some Plasticized PLA/Collagen Bionanocomposites”. We found the comments and suggestions of the reviewers very constructive and we accordingly revised the manuscript. All changes are highlighted in the revised manuscript. Please find attached below this letter, the point-to point answers to the referees.

The Authors of the manuscript entitled In vitro Biocompatibility and Stabilization Effect of AgNPs on Some Plasticized PLA/Collagen Bionanocomposites, submitted to Applied Sciences thank the referee for reviewing our manuscript. We are deeply grateful for the advice and for the observations and comments which we addressed and feel that greatly increased the quality of our manuscript. Please find below the answers to all questions and suggestions.

The paper present an extended full characterization of the subject antimicrobial bio-nanocomposites. It is a useful report but with some not correct evaluation and poor scientific speculation. Some week points needing revision are indicated

1).The objective of the work, last sense of the introduction, is not a motivation but rather a synthetic abstract! Should be implemented indicating the possible importance of the study for knowledge and application.

Answer: Thank you for your suggestion. The objective of the work was improved as follows: “This study could bring important knowledge about the biocompatibility and durability of some bionanocomposites prepared from PLA, collagen and AgNPs with application as coating biomaterial.”

2). UV spectra provide no information. The section could be removed

Answer: The UV spectra section was removed.

3). Normalized FT-IT^R spectra are of poor quality and not adequately resolved to allow determining frequency of absorption

Answer: Figure 2 is shown in a good quality.

4). Figure 4 showing DSC traces. In this last there is no evidence of Tg. Also the type of crystallinity of the sample with 1% Agnps appears much different from all other samples which look similar .WHY ?

Answer: DSC curve for PLA/AgNPs 1% was measured again and showed in Figure 3. All curves are showed clear to observe the thermal transitions.

5).The comments from line 295 to 301 and the fig 7 a)-c) again reports some not clear and probably not correctly corresponding data.

6) with ref to 5) the lines 296-7 report “increasing irradiation dose causes the progress in the process of oxidation happened in PLA/AgNPs 0.5%,”. This is not what fig 7 a indicates

7). Also the lines 298-9 say “... slight irradiation at 10 kGy brings about a diminution of oxidation levels in comparison with the exposure at superior dose of 20 kGy”. Why and how ???Hard to accept.

Answers for 5-7 points: You are right. We change the representation onto more illustrative figures. The new comments are adapted according with these new representations from figure 6:

The nonisothermal CL measurements (Figure 6) are the proofs for the protective effect of AgNPs at different particle concentrations. The upper positions of curves depicting the oxidation of PLA loaded with 0.5 % AgNPs prove the soft stabilization action at this lower concentration. The exposure to different low doses by ɣ-irradiation (10 and 20 kGy) does not alter the sequence of stabilization effects, but it may really assume that the slight degradation of polymer would be accompanied by an oxidation of silver nanoparticles. The increasing concentration of dispersed particles brings about similar level. However, the higher dose determines a faster oxidation of polymer phase on the region of superior temperatures. It means that the progress of oxidation ageing of PLA is less hindered by modified silver nanoparticles when they are subjected to an intense energy transfer. For the explanation of the small difference between the contribution of inorganic component on the delay of oxidation the coalescence of nanoparticles takes place. The availability of these new particle consistence upon the delay of degradation is diminishes and the contribution of higher silver loading can be disregarded. The text was modified accordingly.

8) The discussion is merely descriptive of the results without critics and comparison with supporting literature. For instance the free radical inhibition of Agnps is claimed without confirming support.

Answer: Antioxidant efficacy of synthesized AgNPs was evaluated using 1,1-diphenyl-2-picryl hydrazyl radical (DPPH) [40], H2O2 [41], ABTS radical cation scavenging assay and FRAP assay [42]. The DPPH and ABTS cation radical scavenging activity of AgNPs increased with the increase in the concentration of AgNPs [40,42]. Discussion section was modified accordingly.

9) the concluding section is irrelevant, just a synthetic summary.

Answer: The conclusion section was modified accordingly.

Round 2

Reviewer 1 Report

Journal: Applied Sciences

Title: Development of bionanocomposites based on PLA, collagen and AgNPs and characterization of their stability and in vitro biocompatibility

Authors: Maria Râpă, Laura Mihaela Stefan, Traian Zaharescu, Ana-Maria Seciu, Anca Andreea Èšurcanu, Ecaterina Matei, Andra Mihaela Predescu, Iulian Antoniac, Cristian Predescu

OVERALL COMMENTS/DECISION:

The revised manuscript has been greatly improved. The Introduction section is concise and nicely structured. The M&Ms, Results and Discussion sections have been enriched with related information. I would like to thank the authors for addressing my comments.  However, there are still some weak points in the manuscript in its present form that have to be addressed before its publication.

Major Comment

  1. Page 8, Lines 256-7: ‘It is possible for these components to have a synergetic effect’. The claim for a synergistic effect caused by the inclusion of collagen and AgNPS on the system cannot be supported by the present data. For such a claim, an additional control studying the PLA-AgNPs composite (i.e. without collagen) is needed.

Minor Comments

In the following section, there are some points that must be corrected or rephrased.

General comment: The authors are advised to use a single term, e.g. ‘PLA-based bionanocomposites’, for their developed biomaterials throughout the text.

  1. Page 1, Line 22: Please correct ‘increasing of’ into ‘increase in the’.
  2. Page 1, Line 24: ‘PLA-based material’. Please see General comment and change accordingly.
  3. Page 1, Line 30: ‘PLA bionanocomposites’. Please see General comment and change accordingly.
  4. Page 1, Line 42: Please correct ‘.[1].’ into ‘[1].’
  5. Page 2, Line 47: Please correct ‘and low’ into ‘, low’.
  6. Page 2, Lines 50-52: Please correct ‘By introduction of nanofillers into PLA matrix such as ZnO nanoparticles [9], vinyl POSS (vinyl 50 polyhedral oligomeric silsequioxane) nanoparticles [10], carbon nanotubes [11] and nano-clay [12], another useful features like antimicrobial and UV-light screening properties are also obtained..[1].’ into ‘By introduction of nanofillers, such as ZnO nanoparticles [9], vinyl POSS (vinyl 50 polyhedral oligomeric silsequioxane) nanoparticles [10], carbon nanotubes [11] and nano-clay [12], into PLA matrix another useful feature like antimicrobial and UV-light screening properties are  ’
  7. Page 2, Line 56: Please correct ‘obtained biocompatibility’ into ‘developed biocompatible’.
  8. Page 2, Line 56: Please correct ‘In other paper,[15]’ into ‘In other paper [15],’.
  9. Page 2, Line 61: Please correct ‘favor the cell’ into ‘favor cell’.
  10. Page 2, Line 61: Please correct ‘while the’ into ‘while’.
  11. Page 2, Line 72-73: Please correct ‘to both effective antimicrobial activity and’ into ‘to both effective antimicrobial activity but’.
  12. Page 2, Line 80: Please correct ‘.[23]’ into ‘[23]’.
  13. Page 2, Line 82: Please correct ‘enhancing’ into ‘enhancement’.
  14. Page 2, Line 82: Please correct ‘some PLA’ into ‘PLA’.
  15. Page 2, Line 87: Please correct ‘The paper’ into ‘The present paper’.
  16. Page 3, Line 101: Please correct ‘tendon by National’ into ‘tendon provided by the National’.
  17. Page 3, Line 108: Please correct ‘processing’ into ‘processed’.
  18. Page 3, Line 110: ‘The bionanocomposites’. Please see General comment and change accordingly.
  19. Page 3, Line 121: Please add the details of the company and the country for the mentioned software.
  20. Page 3, Line 123: Please correct ‘distinguished’ into ‘identified’.
  21. Page 3, Line 125: Please correct ‘investigated under’ into ‘characterized using’
  22. Page 4, Line 142: ‘polylactic-based samples’. Please see General comment and change accordingly.
  23. Page 4, Line 154: Please correct ‘5x5 mm discs’ into ‘discs of 5 mm diameter’
  24. Page 4, Line 160: Please correct ‘Finally of’ into ‘At the end of’
  25. Page 4, Line 177: Please correct ‘LDH medium(OD490)/LDH positive control(OD490)’ into ‘(LDH medium(OD490)/LDH positive control(OD490))’
  26. Page 5, Line 192: Please correct ‘10x10 mm discs’ into ‘discs of 10 mm diameter’.
  27. Page 5, Line 194: Please correct ‘phosphate buffer solution (PBS)’ into ‘PBS’
  28. Page 6, Line 208: Please correct ‘is related’ into ‘is shown’.
  29. Page 6, Figure 2: The resolution of the graph is lower compared with the previous version of the manuscript.
  30. Page 7, Line 248: Please correct ‘meaning’ into ‘suggesting’.
  31. Page 8, Line 262: ‘PLA samples’ Please see General comment and change accordingly.
  32. Page 8, Line 269: Please correct ‘is increases’ into ‘increases’.
  33. Page 8, Figure 4: Please correct ‘(c) AgNPs’ into ‘(c) PLA/AgNPs’
  34. Page 8, Line 301: Please correct ‘tested antimicrobial’ into ‘tested’.
  35. Page 11, Figure 8: The captions and scale bars are not visible. Please use different color (e.g. white) for the captions.
  36. Page 13, Line 390: Please correct ‘The introducing’ into ‘The addition’.
  37. Page 13, Line 396: Please correct ‘did no show the increase’ into ‘did not show any increase’.
  38. Page 15, Line 470: Please correct ‘found out’ into ‘found’ or ‘reported’.
  39. Page 15, Lines 471-2: Please correct ‘cytotoxicity antimicrobial’ into ‘cytotoxicity of antimicrobial’.
  40. Page 15, Line 473: Please correct ‘surface-area’ into ‘surface area’.
  41. Page 15, Line 475: Please correct ‘to The Occupational’ into ‘to the Occupational’.
  42. Page 15, Lines 485-7: Please correct ‘The effect of the antimicrobial activity of silver nanoparticles on the prepared bionanocomposites’ into ‘The effect of the silver nanoparticles on the antimicrobial activity of the prepared bionanocomposites’.
  43. Page 15, Line 486: ‘bionanocomposites’ Please see General comment and change accordingly.
  44. Page 15, Line 494-5: Please correct ‘The collagen and AgNPs had a synergetic effect leading to’ into ‘The collagen and AgNPs led to’. Please see major comment#1.

Author Response

Reviewer 1

The Authors of the manuscript entitled Development of bionanocomposites based on PLA, collagen and AgNPs and characterization of their stability and in vitro biocompatibility, submitted to Applied Sciences thank the referee for the overall, major and minor comments addressed. We are deeply grateful for the observations and comments which we addressed and feel that greatly increased the quality of our manuscript. Please find attached the revised manuscript.

 OVERALL COMMENTS/DECISION:

The revised manuscript has been greatly improved. The Introduction section is concise and nicely structured. The M&Ms, Results and Discussion sections have been enriched with related information. I would like to thank the authors for addressing my comments. However, there are still some weak points in the manuscript in its present form that have to be addressed before its publication.

 Major Comment

  1. Page 8, Lines 256-7: ‘It is possible for these components to have a synergetic effect’. The claim for a synergistic effect caused by the inclusion of collagen and AgNPS on the system cannot be supported by the present data. For such a claim, an additional control studying the PLA-AgNPs composite (i.e. without collagen) is needed.

Answer: Thank you for this comment. The phrase “It is possible these components to have a synergetic effect” was deleted.

 Minor Comments

In the following section, there are some points that must be corrected or rephrased.

General comment: The authors are advised to use a single term, e.g. ‘PLA-based bionanocomposites’, for their developed biomaterials throughout the text.

Thank you for your advice, the term “PLA-based bionanocomposites” was introduced into manuscript, where was necessary.

Also, all the minor comments from 1 to 44 points were corrected according your constructive comments.
